# Anxiolytic-like Effects of the Positive GABA_B_ Receptor Modulator GS39783 Correlate with Mice’s Individual Basal Anxiety and Stress Reactivity

**DOI:** 10.3390/ph15020233

**Published:** 2022-02-16

**Authors:** Ahmet Oguzhan Bicakci, Mousumi Sarkar, Yu-Hsin Chang, Evelyn Kahl, Lorenzo Ragazzi, Angel Moldes-Anaya, Markus Fendt

**Affiliations:** 1Integrative Neuroscience Master Program, Institute of Biology, Faculty of Natural Sciences, Otto-von-Guericke University Magdeburg, 39120 Magdeburg, Germany; ahmetoguzhanbicakci@gmail.com (A.O.B.); mousumi.sarkar@st.ovgu.de (M.S.); fiona840703@gmail.com (Y.-H.C.); 2Institute for Pharmacology and Toxicology, Medical Faculty, Otto-von-Guericke University Magdeburg, 39120 Magdeburg, Germany; evelyn.kahl@med.ovgu.de; 3Neurobiology Research Group, Department of Clinical Medicine, UiT The Arctic University of Norway, 9019 Tromsø, Norway; lorenzo.ragazzi@uit.no; 4Nuclear Medicine and Radiation Biology Research Group, Department of Clinical Medicine, UiT The Arctic University of Norway, 9037 Tromsø, Norway; angel.salvador.moldes.anaya@unn.no; 5Cyclotron and Radiochemistry Unit, The PET Imaging Center, University Hospital of North Norway, 9038 Tromsø, Norway; 6Center for Behavioral Brain Sciences, Otto-von-Guericke University Magdeburg, 39120 Magdeburg, Germany

**Keywords:** anxiety, gamma-aminobutyric acid type B receptor, GS39783, light–dark box, stress

## Abstract

Positive gamma-aminobutyric acid type B (GABA_B_) receptor modulators such as GS39783 have showed anxiolytic-like effects in several studies while such effects were absent in other studies. These conflicting findings led us hypothesize that the anxiolytic-like effects of such compounds depend on the individual basal anxiety and/or the anxiogenic properties of the used tests. The present study addresses this hypothesis by testing GS39783 effects on mice’s anxiety-like behavior in a light–dark box. We found that GS39783 had no effects on a whole-group level. However, after grouping the mice for their basal anxiety, GS39783 reduced anxiety-like behavior in the subgroup with highest basal anxiety. Moreover, GS39783 effects correlated with individual basal anxiety. Next, the anxiogenic properties of the light–dark box test were increased by prior stress exposure. Again, GS39783 was not effective on a whole-group level. However, GS39783 had an anxiolytic-like effect in the most stress-responsive subgroup. Moreover, GS39783 effects correlated with individual stress responsiveness. Finally, we show that GS39783 brain levels were within a behaviorally relevant range. Overall, our study demonstrates that GS39783 effects depend on individual basal anxiety and stress responsiveness. This suggests that anxiety tests should generally be designed to capture individual basal anxiety and/or stress responsiveness as well as individual compound effects.

## 1. Introduction

Fear and anxiety are emotions that help animals and humans to cope with threatening situations. However, excessive fear and anxiety are symptoms of several neuropsychiatric disorders, including anxiety and stress-related disorders [1]. Dysfunctions of the brain circuitries mediating emotional responses, particularly fear and anxiety, are thought to be involved in the pathogenesis of anxiety- and stress-related disorders [2,3]. Ultimately, a better understanding of the neuropharmacology of these brain circuitries may enable the development of improved pharmacological therapies for anxiety disorders [4].

Gamma-aminobutyric acid (GABA) is the major inhibitory neurotransmitter in the central nervous system and thus involved in the regulation of many homeostatic physiological and psychological processes [5,6]. Both types of GABA receptors, i.e., ionotropic gamma-aminobutyric acid type A (GABA_A_) receptors and metabotropic gamma-aminobutyric acid type B (GABA_B_) receptors, are abundantly expressed in brain circuitries mediating fear and anxiety, including cortical regions and connected areas such as the amygdala, hippocampus, and hypothalamus [1]. Both preclinical and clinical research indicate that dysfunctions of the GABAergic system are involved in the pathophysiology of anxiety disorders [1,2]. This notion is supported by the potent anxiolytic effects of pharmacological interventions targeting the GABAergic system [7,8]. In clinical practice, benzodiazepines, i.e., positive allosteric GABA_A_ receptor modulators, are mainly used [9,10]. However, benzodiazepines have serious side effects such as muscles weakness, impaired coordination, hangover, amnesia, development of tolerance, and dependence [1,9,11,12]. Because of these side effects, international guidelines recommend benzodiazepines only for short-term treatment, while antidepressants are first choice for long-term treatment of anxiety disorders [13,14]. However, the clinical efficiency of antidepressants has been shown to be limited [15], so there is an unmet medical need for novel anxiolytic treatments.

As an alternative target for anxiolytic treatments, the GABA_B_ receptor has also been explored. GABA_B_ receptor-deficient mice exhibit an anxiogenic-like phenotype [6,16] while treatment with positive GABA_B_ receptor modulators has shown anxiolytic-like effects in several animal paradigms of anxiety, including the light–dark box [17], the elevated plus maze [18,19], the elevated zero maze [17,18,20], stress-induced hyperthermia [18,20,21,22], the staircase test [18,23], and light-enhanced startle [24]. However, positive GABA_B_ receptor modulators did not show anxiolytic-like effects in some of these tests as well as in other tests for anxiety [18,20,24,25,26,27] (see also Appendix A). There are several possible explanations for these variability in the efficacy of positive GABA_B_ receptor modulators: (1) different positive GABA_B_ receptor modulators might have different efficacy due to their binding profile to the receptor; (2) positive GABA_B_ receptor modulators are only effective in particular behavioral paradigms of anxiety but not in others; and/or (3) the anxiety levels in the studies presented here were different and positive GABA_B_ receptor modulators are only effective at particular anxiety levels.

The present study addressed the latter explanation and tested the hypothesis whether the anxiolytic-like effects of a positive GABA_B_ receptor modulator depend on the individual basal anxiety of the mice and/or the anxiogenic properties of the behavioral paradigms. Using the light–dark box paradigm [28] and C57BL/6J mice, we investigated the anxiolytic-like effects of the positive allosteric modulator of the GABA_B_ receptor GS39783 [29]. GS39783 is one of the first discovered positive GABA_B_ receptor modulators and has been used in a variety of different behavioral studies [18,26,30]. In the first experiment, we tested whether anxiety-like behavior, as well as classification of mice according to their anxiety response in the first test, is stable on repeated tests in the light–dark box [31,32]. Based on the positive findings of this first exploratory experiment, we applied a repeated-measure design in the next two experiments, i.e., each mouse was treated with vehicle, 10 and 30 mg/kg GS39783 in a balanced order. In the second experiment, we tested whether anxiolytic-like effects of GS39783 depended on mice’s individual basal anxiety, i.e., the anxiety behavior after vehicle injections. In a third experiment, we enhanced the anxiogenic properties of the test by pre-exposing the mice to stress by electric stimuli and tested whether the anxiolytic-like effects of GS39783 depend on mice’s individual stress responsiveness. Lastly, and to show that the anxiolytic-like effect were directly related to the administration of GS39783, we also measured the plasma and brain levels of the drug in selected satellite animals.

## 2. Results

### 2.1. Experiment 1: Effects of Repeated Testing in the Light–Dark Box

We first tested whether the anxiety-like behavior in the light–dark box test is affected by repeated testing. An analysis of variance (ANOVA) using test day as a within-subject factor revealed that the percent time spent in the bright area of the light–dark box was not affected by repeated testing (F_5,71_ = 0.47, *p* = 0.68; Figure 1A). Very similar results were obtained in the analysis of other behavioral measures in the light–dark box (data not shown) including percent distance travelled in the bright area (F_5,71_ = 0.63, *p* = 0.61), number of entries (F_5,71_ = 1.00, *p* = 0.41) or latency to enter the bright area (F_5,71_ = 0.68, *p* = 0.57). In the following, we focused our analyses on the percent time spent in the bright area of the light–dark box which was the most stable behavioral measure in this study.

In an additional analysis of this experiment, we found a main effect of sex (F_1,10_ = 15.28, *p* = 0.003), i.e., female mice were generally more anxious than male mice. However, post hoc comparisons revealed that this sex difference was most pronounced on test 1 (*t* = 3.92, *p* = 0.04) and did not reach statistical significance on the remaining test days (ts < 2.80, *p* > 0.10). Therefore, sexes were merged in the graphical presentation.

To evaluate whether a classification of mice with respect to their anxiety levels in the first test is stable, mice were grouped into “anxious” and “non-anxious”, based on the behavioral data of the first test (median split). An ANOVA revealed that the two groups showed a robust difference throughout all tests (F_1,10_ = 64.96, *p* < 0.0001, post hoc comparisons ts > 2.55, ps < 0.03; Figure 1B). Again, there was no effect of test day (F_5,50_ = 0.51, *p* = 0.69) and there was no interaction between test day and group (F_5,50_ = 0.29, *p* = 0.92) indicating that a classification of mice based on their anxiety in the first test is very stable. Thereby, an important precondition for the following experiments was fulfilled.

### 2.2. Experiment 2: Effects on GS39783 on Anxiety

Each mouse was treated and tested with vehicle, 10 and 30 mg/kg GS39783 in a balanced order. Figure 2A depicts the percent time spent in the bright compartment which was neither affected by GS39783 treatment (ANOVA: F_2,86_ = 0.30, *p* = 0.75) nor by sex (F_1,43_ = 0.30, *p* = 0.40; interaction treatment x sex: F_2,86_ = 0.11, *p* = 0.90).

To evaluate whether the individual basal anxiety affects the effectiveness of GS39783, we grouped the mice, irrespective of their sex, according to their behavior after vehicle treatment into three groups with the same size (n = 15/group; “anxious”: <31%; “average”: 32–42%; “non-anxious”: >42% time spent in the bright compartment; Figure 2B). Although male mice were a bit over-represented in the non-anxious group, sex did not affect the grouping (Chi^2^ test: χ^2^ = 1.26, *p* = 0.53).

A further ANOVA that now included group as an additional factor revealed a significant interaction between GS39783 treatment and group for the time spent in the bright area (F_4,78_ = 2.81, *p* = 0.03; Figure 2C). There were no interactions of sex with group (F_2,39_ = 0.25, *p* = 0.78) or of sex with treatment and group (F_4,84_ = 1.23, *p* = 0.30). Separated ANOVAs for the different groups showed a significant GS39783 effect in the “anxious” group (F_2,18_ = 3.87, *p* = 0.03) but not in the two other groups (Fs < 1.12, Ps > 0.34). Again, there were no interactions with sex (Fs < 1.45, Ps > 0.25).

Further, the individual effects of 30 mg/kg GS39783 were calculated, i.e., the difference in percent time spent in the bright area after treatment with 30 mg/kg GS39783 and after vehicle treatment. These individual GS39783 effects were well correlated with basal anxiety, i.e., the behavior expressed after vehicle injections (Figure 2D; regression analysis: R^2^ = 0.20, *p* = 0.002). The more anxious mice were after vehicle treatment, the more 30 mg/kg GS39783 increased the time spent in the bright area, i.e., the more anxiolytic-like effects of GS39783 were observed.

### 2.3. Stress-Induced Anxiety in the Light–Dark Box Test

After a light–dark box test without any treatment (Figure 3A; pre-test), mice were exposed to stress by unescapable electric stimuli. Then, 10 days later, they were treated and tested with vehicle, 10 and 30 mg/kg GS39783 in a balanced order (Figure 3A). Stress exposure increased anxiety-like behavior in the light–dark box (comparison pre-test and vehicle); however, this increase did not reach the level of significance (F_1,26_ = 3.68, *p* = 0.07). Sex of the mice had no effects and did not interact with the test session (Fs < 0.26, Ps > 0.62).

In the test sessions after stress exposure, GS39783 treatment had no effects when all mice were analyzed together (Figure 3A; F_2,52_ = 1.79, *p* = 0.18). In addition, sex of the mice had no effects and did not interact with GS39783 treatment (Fs < 0.52, Ps > 0.47). For further analyses, the mice were grouped regarding their stress response (Figure 3B; n = 9–10/group; high, average and low stress responsiveness). Separated analyses of these groups revealed anxiolytic-like effects of GS39783 treatment in those mice which were high responsive to stress (Figure 3C; F_2,16_ = 4.24, *p* = 0.03) but not in those which had average or low stress responsiveness (Fs < 1.38, Ps > 0.27). A correlation analysis further showed that the individual effects of 30 mg/kg GS39783 were well correlated with the individual stress responsiveness (Figure 3D; regression analysis: R^2^ = 0.36, *p* = 0.0007). The more stress responsive mice were, the more 30 mg/kg GS39783 increased the time spent in the bright area. Of note, the individual effects of 30 mg/kg GS39783 did not correlate with the anxiety-like behavior of the pre-test in this experiment (data not shown; regression analysis: R^2^ = 0.0002, *p* = 0.94) but there was a trend for a correlation of the anxiety levels in the pre-test with stress responsiveness (R^2^ = 0.13, *p* = 0.06).

Analysis of the corticosterone plasma levels (Figure 3E) revealed that they were different on the different sampling days (F_5,110_ = 25.80, *p* < 0.0001). However, there were no main effects of sex and stress responsiveness (Fs < 0.17, Ps > 0.85) and there were no interactions between these factors (Fs < 1.31, Ps > 0.26). Post hoc comparison with the baseline corticosterone levels revealed similarly increased corticosterone levels after both the exposure to electric stimuli and the different light–dark box tests (Fs > 18.93, Ps < 0.0001). However, GS39783 treatment had no effects (F_2,44_ = 0.37, *p* = 0.69). Furthermore, corticosterone levels were also not enhanced 9 days after stress exposure, i.e., one day before the light–dark box tests with treatment started (baseline 2; F_1,25_ = 0.95, *p* = 0.34).

### 2.4. Plasma and Brain Levels of GS39783

GS39783 was found in plasma samples and brain tissue extracts from the treated satellite animals (Figure 4). Plasma and brain levels of these mice that received 30 mg/kg GS39783 are shown in Table 1. The mean concentrations of GS39783 in the plasma was 217 µg/mL while for the brain tissue the mean concentration was found to be 57 µg/g. The brain-to-plasma ration is approximately 0.26 indicating a significant brain penetration of GS39783.

## 3. Discussion

The present study investigated the effects of the positive GABA_B_ receptor modulator GS39783 on anxiety-like behavior of mice in the light–dark box test. First, we showed that anxiety-like behavior and classification of the mice in the first test is stable in repeated tests in the light–dark box. This finding enabled us to use a repeated-measure design for the treatments with vehicle, 10 and 30 mg/kg GS39783. Neither GS39783 dose had significant effects on mice’s anxiety-like behavior when analyzed at the whole-group level. However, when the mice were grouped according to their basal anxiety, i.e., after vehicle administration, GS39783 had significant anxiolytic-like effects in the higher anxiety group but not in the groups with average and low anxiety. Comparable observations were made in another experiment in which mice were first exposed to stress by electric stimuli and 10 days later were tested for GS39783 effects in the light–dark box test. GS39783 had anxiolytic-like effects in the mice that responded with an increase in anxiety after the stress, but not in those in which the stress did not alter anxiety. Last, our analyses of plasma and brain levels showed significant brain penetration of GS39783, indicating a secrete potency in exerting a pharmacological response.

When the first positive GABA_B_ receptor modulators were developed [29], it was hoped that these compounds would have the desired effects of the positive GABA_A_ receptor modulators (benzodiazepines) but without their undesirable side effects [33]. Indeed, many studies showed anxiolytic-like properties of GS39783, one of the first positive GABA_B_ receptor modulators, in a variety of anxiety tests in laboratory rodents, while some of the side-effects of benzodiazepines or the GABA_B_ receptor agonist baclofen, such as sedation, hypothermia, interaction with ethanol, or development of tolerance, were not observed with GS39783 [17,18]. Very similar results were obtained with other positive GABAB receptor modulators such as CGP7930 [20], ADX1441 [19] or BHF117 [24,27]. However, in some anxiety tests, the positive GABA_B_ receptor modulators had no effects on anxiety-like behavior [18,20,24,25,26,27].

There are several possible explanations for these conflicting results. First, different tests for anxiety might show different sensitivity to anxiolytic-like effects. It is also plausible that the different positive GABA_B_ receptor modulators have different potency when interacting with the receptor. Another explanation could be that positive GABA_B_ receptor modulators can exert an anxiolytic-like effect only if animals have high anxiety and/or stress levels. That would imply that the anxiolytic-like effects of positive GABA_B_ receptor modulators depend on the basal anxiety levels of the animals and/or the anxiogenic properties of the different anxiety tests. This explanation is supported by the observation that positive GABA_B_ receptor modulators had anxiolytic-like effects in studies in which high anxiety levels were measured in the control groups (after vehicle injections) but were ineffective in studies in which medium/low anxiety levels were observed. In similar studies, anxiolytic-like effects were seen on the elevated plus maze when vehicle-treated animals spent approximately 5–8% of the time on the open arms, i.e., at high basal anxiety [18,19], but not when vehicle-treated mice spent approximately 25–30% of the time on the open arms, i.e., at middle/low basal anxiety [24,25] (see also Appendix A). These observations led us to hypothesize that the compounds’ anxiolytic-like effects depend on the basal anxiety of each individual animal in a determined anxiety test [34].

There are mainly two experimental approaches to investigate this hypothesis. Either animals with different basal anxiety levels are used, or the anxiogenic properties of the behavioral test are modified. In the present study, we used both approaches. In our second experiment, we grouped the mice according to their basal anxiety levels, i.e., their anxiety-like behavior after vehicle injections. In the third experiment, we increased the anxiogenic properties of the test by exposing the mice to stress by electric stimuli 10 days before the test and grouped the mice according to their stress responsiveness, i.e., the stress-induced increase of anxiety. For both approaches, it was necessary to test the mice multiple times, i.e., both after vehicle and after GS39783 treatment. However, such multiple testing is not advisable for every anxiety test. For example, anxiety-like behaviors in the elevated plus maze test change a lot with repeated testing [35,36]. In contrast, mice’s behaviors in the light–dark box test appear to be relatively stable [31,32], making this test an optimal candidate for the present study.

The aim of our first experiment was therefore to confirm the stability of mice’s behaviors in the light–dark box test. The present data showed no effects of repeated testing, not only at the level of the whole group’s mean (Figure 1A), but also at a subgroup level, i.e., when the mice were subgrouped according to their anxiety levels in the first test (Figure 1B). Based on these findings, the following two experiments addressed the questions whether the anxiolytic-like effects of GS39783 treatment depended on mice’s basal anxiety or on anxiogenic properties of the test protocol. In both experiments, mice were repeatedly treated and tested in the light–dark box experiment. The following approaches were used to analyze possible mean and individual anxiolytic-like effects of GS39783 [37]: (1) all tested mice were analyzed together; (2) The mice were divided into three subgroups according to their basal anxiety levels after vehicle injections (experiment 2) or according to their change in anxiety levels after exposure to stress (experiment 3). We then analyzed whether GS39783 has anxiolytic-like effects in the different subgroups; (3) We checked whether individual basal anxiety or stress responsiveness, respectively, was correlated with the effects of GS39783.

The different analytical approaches confirmed our hypothesis. When the whole group was analyzed, GS39783 treatment had no significant anxiolytic-like effects (Figure 2A and Figure 3A). After subgrouping the mice according to basal anxiety or stress responsiveness, respectively, anxiolytic-like GS39783 effects were observed in the most anxious or most stress-responsive groups, respectively (Figure 2C and Figure 3C). Last, the correlation analyses showed that the more anxious or the more stress-responsive individual mice were, the stronger the anxiolytic-like GS39783 effects were (Figure 2D and Figure 3D).

What are the conclusions from this study? Our findings show that anxiolytic-like effects are best observed with high basal anxiety or high anxiogenic properties of the test paradigm. Therefore, it might be best to use either animals with high basal anxiety or a test paradigm with high anxiogenic conditions. The former should be achieved, for example, by using a strain with high basal anxiety [38] or animals selectively bred for high anxiety [39]; the latter could be achieved, for example, by adjusting illumination [40] or other test parameters, or by initially exposing animals to stress [41]. Without doubt, such approaches would be the simplest ways to detect anxiolytic-like properties of a test substance.

However, we believe that test substances can be better characterized by using animals and test conditions that allow subgroups to be formed according to their basal anxiety or stress responsiveness. These subgroups reflect the variability in anxiety or stress responsiveness that is observed at the population level in animals as well as in humans. From a translational perspective, anxiolytic-like medications should reduce pathologically-increased anxiety to a healthy, i.e., adaptive levels in patients with anxiety or stress-related disorder. However, these medications should not reduce healthy, i.e., adaptive, anxiety levels as this could lead to risk-prone and thus maladaptive behaviors. Based on this requirement, GS39783 shows this desired profile in the present experiments. It did not reduce low or average anxiety levels, which are adaptive, but only high anxiety levels, which are maladaptive.

Although we found positive effects of GS39783 at the behavioral level in the most anxious or stress-responsive mice, their hormonal stress response, i.e., corticosterone plasma levels, was not affected by GS39783 after the light–dark box test (Figure 3E). This was also the case when mice were grouped according to their acute hormonal stress response (data not shown). Notably, corticosterone plasma levels after the exposure to electric stimuli were very similar to those measured after the light–dark box tests (including intraperitoneal injections), suggesting a similar, rather mild stressfulness of these two different interventions. To our knowledge, there are no published studies that measured the corticosterone levels after treatment with GS39783 or other positive GABA_B_ receptor modulators. However, GABA_B_ receptor deficiency is not associated with changes in corticosterone levels [42]. The present data supports this finding.

Last, we also present first pharmacokinetic data on GS39783 in some satellite animals treated with 30 mg/kg GS39783. Mean brain levels of approximately 57 µg/mg were measured, corresponding to approximately 170 µM. This is a concentration that significantly potentiates GABAergic activity in recombinant cell assays [29]. Of note, oral administration of 100 mg/kg GS39783 was able to potentiate GABA_B_-mediated effects in vivo only when the GABA_B_ receptor antagonist baclofen was additionally applied [30]. This effect was not observed with oral administration of lower doses of GS39783 [30], although behavioral effects were observed as early as 10 mg/kg GS39783 [18]. In the present study, behavioral effects were observed only after intraperitoneal administration of 30 mg/kg GS39783, but not at 10 mg/kg GS39783. This may indicate that bioavailability of GS39783 is higher after oral administration than after intraperitoneal administration. Further experiments are required to fully understand the pharmacokinetic properties of GS39783.

## 4. Materials and Methods

### 4.1. Animals

Experimentally naive male and female C57BL/6J mice (in-house colony; breeding pairs from Charles River Laboratories, Sulzfeld, Germany) were used in the present study. Animals were 2–3 months old during the experiments. Each cage housed four to six animals. Housing conditions were maintained during the duration of the study with controlled humidity (50–55%), temperature (22 ± 2 °C) and 12 h/12 h light/dark cycle (lights-on: 6:00 a.m.). All animals had access to food and water ad libitum. The experiments were conducted in the lights-on phase between 9:00 a.m. and 3:00 p.m. and animals were weighed before each experiment. The European Union regulations for animal care and use for experiments were strictly adhered in all experiments (2010/63/EU). Moreover, ethical approval was taken from the local authorities of the State of Saxony-Anhalt, Germany (Landesverwaltungsamt Sachsen-Anhalt, 42502-3-747 and 42502-2-1172 Uni MD).

### 4.2. Chemical Reagents

GS39783, a positive allosteric GABA-B receptor modulator was purchased from Axon MedChem BV (ID: 1820; Groningen, The Netherlands), Tween-80 was purchased from Sigma-Aldrich Chemie (P5188; Taufkirchen, Germany). Acetonitrile and formic acid were purchased from VWR, both LC-MS quality (#1.00029.1000 and #85048.001, Oslo, Norway). Water was purified by in a Milli-Q IQ 7000 Ultrapure Water System (Merck-Millipore, Oslo, Norway).

### 4.3. Compound Formulation for Injection

GS39783 was formulated in 1% Tween-80 in saline; 30 min before the respective experiments, vehicle (control), 10 or 30 mg/kg GS39783 were injected intraperitoneally with a dose volume of 10 mL/kg.

### 4.4. Behavioral Experiments

Four boxes (49.5 cm × 49.5 cm × 41.5 cm), each of them separated into one bright and one dark compartment, were used in the experiment. The bright compartments had an illumination between 290–320 lux and the dark compartments 1–2 lux. The size of the compartments were equal and they were connected by an opening (8 cm × 6 cm). The boxes were equipped with infrared detector frames (sensor distance: 16 mm) allowing the tracking of the animals’ movements (TSE Systems, Bad Homburg, Germany). For the tests, the mice were put into the dark compartment of the light–dark box and test duration was 10 min. As behavioral measures of anxiety, the latency to enter the bright compartment, the percent time spent in each of the compartments, the distance travelled in the two compartments, and the number of transitions between two compartments were used.

Three different behavioral experiments were performed in the light–dark boxes (Figure 5):

Experiment 1: The aim of this experiment was to test whether anxiety-like behavior in the light–dark box as well as classification of the mice according to their anxiety behavior in the first test was stable during repeated testing. Therefore, 6 male and 6 female mice were exposed six times to the light–dark box, with breaks of 2–3 days between each test session.

Experiment 2: The aim of the second experiment was to test the effects of GS39783 on anxiety-like behavior in the light–dark box. In total, 25 male and 20 female mice were used. Each mouse was tested three times with a break of 2 days between each test. Mice received vehicle, 10 and 30 mg/kg GS39783 in a balanced order (latin-square design) and 30 min before the tests.

Experiment 3: The third experiment tested GS39783 effects on stress-potentiated anxiety-like behavior in the light–dark box. First, 16 male and 12 female mice were tested in the light–dark box without any treatment. One day later, all mice—again untreated—were exposed to five aversive electric stimuli (1 s duration, 0.7 mA intensity, 60 to 150 s inter-stimulus interval) in a startle system (SR-LAB, San Diego Instruments, San Diego, CA, USA). Next, 10 days later, the animals were again tested in the light–dark box as in experiment 2, i.e., the three different treatments in a balanced order and a break of 2 days between each of the tests.

To analyze plasma corticosterone levels, six blood samples were collected from each mouse by tail vein cutting on the following days: (1) one day before the first light–dark box test; (2) 30 min after exposure to electric stimuli; (3) one day before starting the light–dark box tests with treatments; and (4–6) 30 min after each light–dark box test. Corticosterone levels were quantified in 100 times diluted plasma samples by an ELISA kit (ADI-901-097; Enzo Life Sciences, Lörrach, Germany), which was performed according to the manufacturer’s guidelines.

**Figure 5 pharmaceuticals-15-00233-f005:**
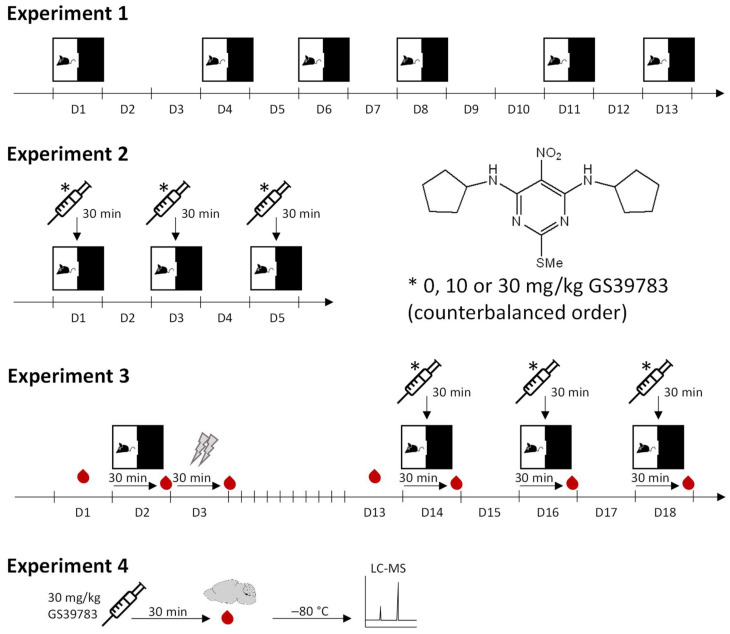
Timelines of the different experiments. The squares with the mouse symbolize the light–dark box test (10 min test duration), the syringes symbolize the intraperitoneal injections, the red drops symbolize the blood collections by tail cutting (in experiment 4: trunk blood), and the flashes symbolize the stress by exposure to electric stimuli. Abbreviations: D, day; LC-MS, liquid chromatography-mass spectrometry.

### 4.5. Identification and Semi-Quantitative Analysis of GS39783 in Plasma and Brain (Experiment 4)

To determine plasma and brain levels of the administered GS39783, three experi-mentally naive male mice were treated with 30 mg/kg of the compound as described in Section 4.3. Subsequently, 30 min post-administration, the animals were sacrificed and trunk blood was collected and brains were dissected out of the animals. The brains were washed with isotonic saline, and all samples were stored at −80 °C until further analysis (Figure 5).

On the day of the analysis, brain tissue was homogenized with isotonic saline and extracted with ice-cold acetonitrile (*v*/*v;* 1:1). Plasma samples were extracted directly with ice-cold acetonitrile (1:1) and both tissue and plasma samples were centrifuged (12,000× *g*, 10 min at 4 °C) in a refrigerated centrifuge (Mega Star 1.6R, VWR, Oslo, Norway). Particle-free samples were subsequently analyzed by a Quadrupole Time-of-Flight (QToF) liquid chromatography–mass spectrometry (LC-MS) instrument (6530 model, Agilent, Waldbronn, Germany) using a water/acetonitrile mobile phase modified with 0.1% formic acid and applying a gradient over 10 min. GS39783 was eluted from a C18 BEH (1.7 µm, 50 × 2.1 mm) reverse-phase ultra-high-performance liquid chromatography (UHPLC) column (Waters, Milford, MA, USA) after approximated 6.5 min. The method is pre-validated according to ICH Q2(R1) guidelines from the European Medicine Agency (EMA, www.ema.europa.eu, accessed on 15 May 2021).

### 4.6. Descriptive and Analytical Statistics

For the statistical analysis of the data, GraphPad Prism (version 8; GraphPad Software Inc., San Diego, CA, USA) and Systat (version 13; Systat Software Inc., Chicago, IL, USA) were used. Data are shown in means ± SEM (standard error of the mean). For statistical analysis, normal distribution was first confirmed with D’Agostino–Pearson test. Then, analyses of variance (ANOVA) was performed followed by Sydak’s multiple comparison tests. Furthermore, linear regression analyses were performed.

## 5. Conclusions

Taken together, the present study shows that the anxiolytic-like effects of the positive GABA_B_ receptor modulator GS39783 depend on the individual basal anxiety and/or stress responsiveness of the mice. The more anxious or stress-responsive the mice were, the more pronounced were GS39783’s effects, whereas no effects were observed in non-anxious mice. We believe these observations are not specific to GS39783 or to positive GABA_B_ receptor modulators but may be a feature of many drugs with potential anxiolytic-like properties. Ultimately, the pharmacodynamic profile observed in the present study is desired for anxiolytic-like drugs. They should treat high anxiety states but not reduce anxiety in healthy subjects. This profile would have been overlooked if only the classical group comparison approach had been used in the present study. Therefore, the present study also advocates the use of protocols that allow the determination of basal anxiety or stress responsiveness and the association of these individual characteristics with compounds’ anxiolytic-like effects. Such protocols provide a more detailed understanding of already known and novel compounds with potential anxiolytic-like effects and reduce the probability that anxiolytic-like effects will be overlooked. In the end, patients could also benefit if drugs with better anxiolytic properties are tested in clinical trials as a result of improved preclinical characterization.

## Figures and Tables

**Figure 1 pharmaceuticals-15-00233-f001:**
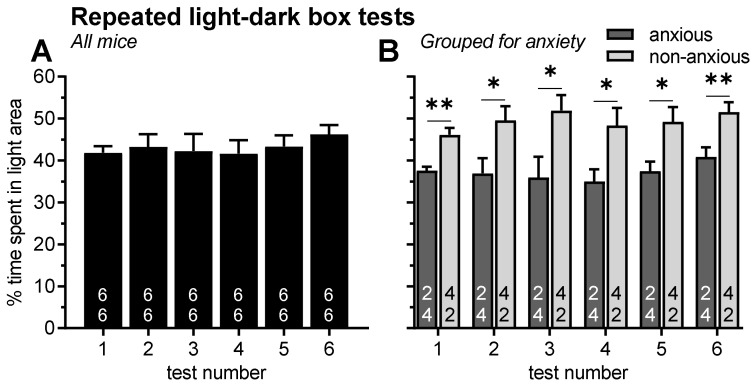
Diagrams depicting anxiety-like behavior in the light–dark box during repeated tests. There were no effects of test day, neither when all mice were analyzed together (**A**), nor when mice were grouped in anxious and non-anxious animals based on their behavior in the first test (**B**). ** *p* < 0.01, * *p* < 0.05, comparisons as indicated. Numbers in the bars represent group sizes for males (top) and females (bottom). Y-axis scale and units in panel (**B**) are the same as in panel (**A**).

**Figure 2 pharmaceuticals-15-00233-f002:**
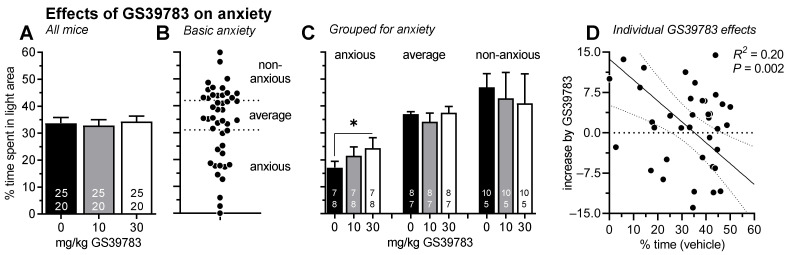
Diagrams showing the effects of 10 and 30 mg/kg GS39783 on the percent time spent in the bright area of the light–dark box. There were no effects of GS39783 if all animals were analyzed together (**A**). For further analysis, mice were then grouped based on their basal anxiety, i.e., their behavior after vehicle (**B**). If mice were grouped for basal anxiety, 30 mg/kg GS39783 had anxiolytic-like effects in the mice of the most anxious group but not in the average and non-anxious groups (**C**). Furthermore, the individual effects of 30 mg/kg GS39783 were correlated with the behavior after vehicle (**D**). * *p* < 0.05, comparisons as indicated. Sexes were pooled since there were no effects of sex (numbers in the bars represent group sizes for males (top) and females (bottom)). Y-axis scale and units in panels (**B**,**C**) are the same as in panel (**A**).

**Figure 3 pharmaceuticals-15-00233-f003:**
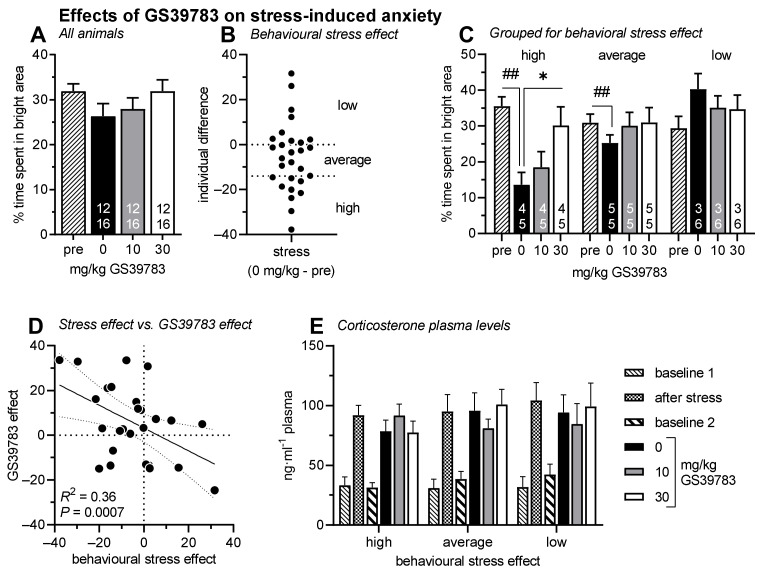
Diagrams showing the effects of 10 and 30 mg/kg GS39783 on the percent time spent in the bright area of the light–dark box and on corticosterone plasma levels 10 days after stress by exposure to electric stimuli. There were no main effects of GS39783 treatment (**A**). However, the change of light–dark box behavior after stress was very variable (**B**). Therefore, mice were grouped according to their stress response (**C**). In the high stress-responsive group, GS39783 treatment significantly increased percent time spent in the bright area of the light–dark box. GS39783 treatment had no effects in the groups with average or low stress responsiveness. Of note, individual stress responsiveness was correlated with the effects of 30 mg/kg GS39783 (**D**). Corticosterone plasma levels were increased after the stress as well as after the light–dark box tests but were not affected by stress responsiveness and treatment (**E**). ## *p* < 0.01, * *p* < 0.05, comparisons as indicated. Sexes were pooled since there were no effects of sex (numbers in the bars represent group sizes for males (top) and females (bottom)).

**Figure 4 pharmaceuticals-15-00233-f004:**
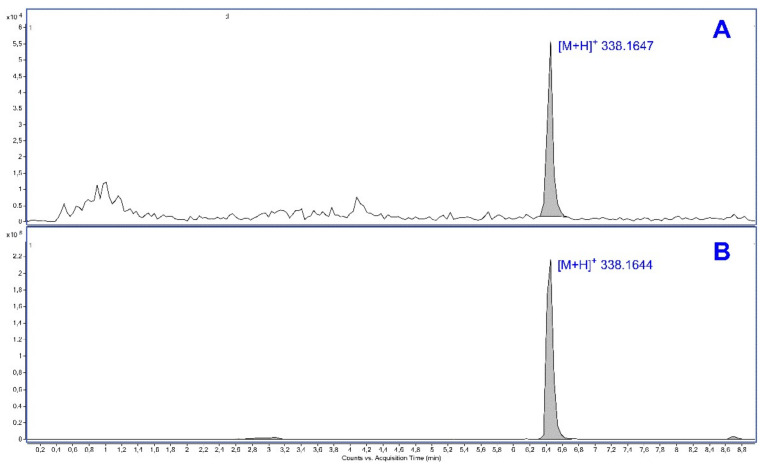
Extracted ion chromatograms of mass-to-charge ratio (*m*/*z*) 338.1640 corresponding to [M+H]^+^ ion of GS39783 in positive mode electrospray (ESI) from (**A**) mouse brain extract and (**B**) plasma sample. Peak areas were used for quantitative purposes. Note the values measured in the peak apex (*m*/*z* 338.1647 and *m*/*z* 338.1644).

**Table 1 pharmaceuticals-15-00233-t001:** Summary of the GS39783 levels found in plasma and brain tissue of selected animals administered with an intraperitoneal dose of 30 mg/kg.

Mouse	Plasma (µg/mL)	Brain (µg/g)	Brain/Plasma Ratio
1	219.8	71.0	0.32
2	222.3	74.2	0.33
3	209.8	26.1	0.12
Mean ± SEM	217.3 ± 3.8	57.1 ± 15.5	0.26 ± 0.07

## Data Availability

Data is contained within the article and Appendix A.

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
