# Peer review of "Anxiolytic-like Effects of the Positive GABAB Receptor Modulator GS39783 Correlate with Mice’s Individual Basal Anxiety and Stress Reactivity"

_pharmaceuticals, 2022, doi:10.3390/ph15020233_

Round 1

Reviewer 1 Report

This work studied the anxiolytic-like effects of the positive GABAB receptor modulator GS39783 and its correlation with mice’s individual basal anxiety and stress reactivity. There are some concerns in this manuscript as follows:

  • Abstract:

- The meaning of the abbreviations should be clearly defined at their first mention (e.g. GABAB).

- Key words: The meaning of “GABAB” should be clearly mentioned not abbreviated.

  • Introduction:
  1. The novel points in this study should be clarified because there are previous studies that suggested the hypothesis that the response to anxiolytics is variable according to the basal anxiety level of the individual.
  2. The reason for selecting GABAB receptor modulator GS39783 in this study among other anxiolytics should be addressed in the “Introduction” section.
  • Materials and methods:
  1. The exact source, concentrations and the catalogue numbers of the used drugs, kits and chemicals should be mentioned.
  2. How did you know that the animals were acclimatized?
  3. In statistical analysis, which version of GraphPad and Systat were used?
  • Results:

I think that it will be better to express the results as mean ± standard deviation to clearly delineate the level of significance.

  • Discussion:

The discussion should be summarized to focus on analysis of the results of the present study.

  • Conclusion:

- I think that the conclusion was not sufficient. Suggestion of further experiments to evaluate the possible clinical implications of the results of the present study should be clearly addressed.

  • General comments:
  1. The manuscript should be revised to improve the quality of the language.
  2. The meaning of the abbreviations should be clearly defined at their first mention.

Author Response

Thanks a lot for your helpful comments.

For our answers, please see the attachment.

Reviewer 2 Report

A discrepancy of positive GABAB receptor modulators like GS39783 in anxiolytic-like effects prompted the authors to hypothesize that the anxiolytic-like effects of such compounds are dependent on individual basal anxiety and/or the anxiogenic properties of the test. The hypothesis was tested in three different behavioral experiments in the light-dark boxes. Additionally, the corticosterone levels were measured after treatment with GS39783 or other positive GABAB receptor modulators. Moreover, semi-quantitative analysis of GS39783 in plasma and brain was performed.

The present study is a very solid piece of work (methodology for testability of the hypothesis, experiment control, etc.).

However, the Reviewer wants to know what is an important message for behavioral anxiolytic-like studies? Is the observed effect true only for the positive GABAB receptor modulator or for all “weak” anxiolytics with a different mechanism of action? Maybe the lack of basal anxiety or stress responsiveness determination caused the deletion of new compounds from further preclinical research? It should be discussed in the text.

Minor:

  1. "their behavior after 0 mg/kg GS39783" – dose of 0 mg/kg looks ridiculously. Vehicle?
  2. "Analysis of the corticosterone plasma levels (Figure 3E) revealed that there was an effect of the sampling day (…)" - It was an effect of a weekday? Please clarify.

Author Response

Thanks a lot for your comments.

For our answers, please see the attachment.
